# Neoadjuvant Stereotactic Ablative Radiotherapy in Pancreatic Ductal Adenocarcinoma: A Review of Perioperative and Long-Term Outcomes

**DOI:** 10.3390/diseases13070214

**Published:** 2025-07-08

**Authors:** Robert Michael O’Connell, Emir Hoti

**Affiliations:** Department of Hepatopancreaticobiliary and Transplant Surgery, St Vincent’s University Hospital, Elm Park, Dublin 4, D04 W6N7 Dublin, Ireland

**Keywords:** pancreaticoduodenectomy, stereotactic ablative radiotherapy, pancreatic ductal adenocarcinoma

## Abstract

The incidence of pancreatic ductal adenocarcinoma (PDAC) is continuing to rise globally, while overall survival continues to be poor. Margin-negative (R0) surgical resection is essential to improve patient outcomes. With increasing understanding of the importance of anatomy and biology to establishing the resectability of PDAC, neoadjuvant therapy (NAT) has emerged as an important strategy to achieve an R0 resection, particularly for those with borderline resectable (BR-PDAC) and locally advanced disease (LA-PDAC). However, despite the multiple randomised controlled trials (RCTs) published in recent years, the optimum regime has yet to be fully established. The role of neoadjuvant chemoradiation therapy (CRT) remains controversial, possibly allowing for improved local disease control at a potential cost of interrupting systemic treatment. The emergence of stereotactic ablative radiotherapy (SABR), in place of conventional radiation therapy, improves patient tolerance of NAT and may improve local tumour control for patients with PDAC during limited fractions, minimising systemic therapy interruption. A particular niche for SABR may be as part of NAT for LA-PDAC, potentially converting a minority of patients with favourable biology to allow for resection. While pancreaticoduodenectomy can be technically challenging following NAT, there is no difference in the rate of major morbidity or mortality post operatively. Indeed, post-operative pancreatic fistula (POPF) rates may be lower following NAT. Overall, however, evidence for SABR in a neoadjuvant setting for BR- and LA-PDAC remains sparse.

## 1. Introduction

Over 400,000 new cases of pancreatic ductal adenocarcinoma (PDAC) are diagnosed globally each year, with the highest incidence seen in Europe and North America [1]. Despite advances in care in recent decades overall 5-year survival remains below 5%; however, for those patients with surgically resected disease, the 5-year survival is approximately 20% [2]. Unfortunately, only 10% of patients present with early-stage disease, meaning upfront surgery is not an option for the majority of patients with PDAC [3].

### Epidemiology of PDAC

The incidence of PDAC continues to rise, with the incidence in younger people, especially younger women, rising the fastest [4,5]. The principle risk factors for developing PDAC can be considered to be modifiable, in particular obesity, smoking, and alcohol consumption, and non-modifiable, such as hereditary factors [6]. Only approximately 10% of cases of PDAC occur in the context of known hereditary or genetic risk factors, while 90% are sporadic [7]. A number of genetic risk factors have been described, including hereditary breast and ovarian cancer syndrome (BRAC1, BRCA 2, and PALB2 genes), Lynch syndrome (MLH1, MSH2, MSH6, PMS2, EPCAM genes), familial adenomatous polyposis (APC gene), Peutz-Jeghers syndrome (STK11, LKB1 genes), hereditary pancreatitis (PRSS1, SPINK 1), and cystic fibrosis (CFTR gene) among others, which increase the incidence of PDAC by between 2 and 132 times [8]. Familial pancreatic cancer, which is a family clustering of PDAC with at least two affected first-degree relatives (FDRs) without an identified hereditary cancer syndrome, accounts for the majority of cases of hereditary pancreas cancer [9]. The relative risk (RR) of developing PDAC is 6.4 times for people with two affected FDRs, and with three FDRs the RR is 32.0 [10].

As noted above, the majority of cases of PDAC are sporadic and a number of important modifiable risk factors exist. Obesity represents a significant public health challenge, with nearly 60% of adults in Europe overweight or obese [11]. Obesity increases the risk of developing PDAC by 1.5 times, and the risk increases with increasing BMI and a younger age of onset of obesity [12]. The relationship between obesity and PDAC is multifactorial, with obesity associated chronic inflammation, insulin resistance and metabolic dysfunction, relative immunosuppression, and altered production of hormones such as adiponectin, ghrelin, and leptin by adipose tissue thought to be contributing factors [13]. Closely linked to obesity, type 2 diabetes (T2DM) is associated with a 1.5–2.0 relative risk of developing PDAC, but T2DM may also be precipitated by the development of PDAC [14]. Interestingly, long-term follow-up data for patients undergoing bariatric surgery for obesity show a significant reduction of the risk of developing PDAC (HR 0.567), particularly in younger patients or those with pre-existing T2DM [15,16].

Smoking represents a significant modifiable risk factor for PDAC, with current smokers having a RR of 1.66 (CI 1.38–1.98) compared to non-smokers and former smokers having a RR of 1.4 (CI 1.16–1.67) compared to never smokers [17]. Tverdal et al. showed a dose-dependent link between alcohol consumption and the risk of PDAC, with an HR of 1.08 (CI 1.02–1.15) for men and 1.04 (0.97–1.13) per 1 unit of alcohol per day, although this is confounded by smoking status [18]. A separate meta-analysis showed heavy drinkers had a RR of 1.29 (CI 1.2–1.38), while light drinkers had a RR of 0.96 (CI 0.75–1.22) compared to never drinkers [17].

Despite the increasing incidence of PDAC, patient outcomes have been slow to improve and remain poor [19]. This is in part because most patients present with either metastatic or locally advanced disease [3]. A key focus in the management of PDAC is identifying those patients who may be candidates for surgical resection, with or without neoadjuvant therapy (NAT), and therefore the concept of “resectability” is important in PDAC [20].

## 2. Defining Resectability in PDAC

The goal of surgery in PDAC should be to ensure a negative resection margin (R0) where possible [21]. The lack of a universally agreed criteria as to what constitutes an R1 resection limits the comparability of data, but reported post-pancreaticoduodenectomy R1 rates range from 10 to 90% in the literature [22]. Achieving an R0 resection is of critical importance: the median overall survival (OS) after R0 resection can be double that after an R1 resection (41.0 vs. 20.6 months *p* = 0.002), while R1 is an independent risk factor for reduced overall survival (HR 1.56, CI 1.07–2.26) [23].

There is no universally agreed definition of borderline resectable PDAC (BR-PDAC); however, as a broad concept it represents approximately 10% of patients with PDAC who have disease that may technically resectable but who have a significantly increased risk of positive resection margins (R1) and therefore local recurrence [24]. An anatomical definition of BR-PDAC was initially adopted by the National Comprehensive Cancer Network (NCCN) in 2006 and has been updated several times since. Currently, tumours in the head/uncinate process of the pancreas considered to be BR-PDAC include: those with a solid tumour contact of 180° or more to the superior mesenteric vein SMV) or portal vein (PV), or those with distortion or thrombosis of the vein but suitable for resection and reconstruction; those with contact of less than 180° with the superior mesenteric artery (SMA); and those with contact with common hepatic artery (CHA) without extension to coeliac axis (CA) or CHA bifurcation suitable for resection and reconstruction [25]. In the body or tail of the pancreas, the NCCN definition considers tumours with up to 180° of contact with the CA, or encasement of the CA without involvement of the aorta or gastroduodenal artery (GDA) to be borderline resectable. The NCCN definition is not universally accepted, and other anatomical definitions such as the MD Anderson definition and the definition used in the Alliance A021101 trial exist [26].

While BR-PDAC traditionally relied on anatomical definitions of resectability, there is growing evidence that patients with evidence of more aggressive tumour behaviour should also be classified as having BR-PDAC [27]. While this is not factored into the current NCCN criteria, the 2016 International Association of Pancreatology (IAP) consensus statement on BR-PDAC included a biological definition of BR-PDAC (suspicion of, but no proven, distal metastasis; serum CA19.9 > 500 u/mL; regional lymph nodes on biopsy or PET-CT) and a conditional definition of BR-PDAC (patients with an ECOG performance status of 2 or more) in those with anatomically resectable PDAC [25]. Certainly, lymph node positivity has been shown to be correlated with poorer OS and reduced disease-free survival (DFS) [28]. Subsequent validation of the CA19.9 > 500 u/mL cut off proposed by the IAP has shown mixed results: Kato et al., for example, showed that a cut off of >1000 u/mL rather than >500 u/mL was predictive of OS [29]. A post hoc analysis of the Dutch PREOPANC-1 and Korean NAT randomised controlled trials (RCTs) failed to show a difference in treatment effect for those with Ca19.9 above and below the 500 u/mL cut off, but those with Ca19.9 < 500 u/mL at presentation were more likely to have a treatment response to NAT [30].

Approximately 30–40% of patients present with locally advanced PDAC (LA-PDAC) that is not metastatic but considered to be not resectable at the time of presentation due to extensive vascular involvement [31]. As with BR-PDAC, there is no universally agreed definition of LA-PDAC. The NCCN definition classes those tumours with >180° involvement of the SMA or CA, or <180° involvement of CA with aortic involvement, or with >180° involvement of SMV or PV or contour irregularity or thrombosis where reconstruction is not possible. Local involvement of the proximal jejunal tributaries of the SMV is often considered to represent unreconstructable disease, not least because of the significant anatomical variation in the venous anatomy here [32]. Ultimately, less than 30% of patients with LA-PDAC will ultimately undergo resection, with median survival reported to be from 8.9 to 25 months [33].

## 3. The Evidence for Neoadjuvant Therapy in BR-PDAC

NAT for BR-PDAC was first proposed in the 1990s in MD Anderson in the United States, where chemoradiotherapy (CRT) involving fluorouracil and 50.4 gray (Gy) external-beam radiation therapy (EBRT) were administered to patients with BR-PDAC [34]. Multiple RCTs have shown a survival benefit with NAT followed by surgery rather than upfront surgery and adjuvant treatment for patients with BR-PDAC [35]. However, the optimum regime for NAT is still the subject of debate, with FOLFIRINOX now being the most commonly used chemotherapy regimen [36]. This may be combined with CRT involving long-course EBRT and oral capecitabine in order to improve local tumour control [37]. Concerns do, however, remain about treatment toxicity with NAT. The NORPACT-1 trial, for example, which evaluated the role of NAT with FOLFIRINOX compared to upfront surgery in patients with resectable PDAC, found that 58% of patients receiving NAT experienced at least one grade 3 or higher toxicity, which may impact patients’ fitness for surgery [38].

In 2018, Jang et al. published the results of a Korean open-label phase II/III RCT of 58 patients with BR-PDAC (using NCCN 2012 criteria) who were randomised to either the NAT (CRT involving 45 Gy in 25 fractions and 9 Gy in 5 fractions (5 times/week for 6 weeks) plus intravenous (IV) gemcitabine) with a 4–6 week break before restaging and then surgery arm or the upfront surgery arm [39]. In the upfront surgery group, patients underwent adjuvant CRT using the same protocol as the NAT group. Both groups were then given four cycles of adjuvant gemcitabine. In the intention to treat (ITT) analysis, 2-year OS and median survival were better in the NAT group (40.7% and 21 months vs. 26.1% and 12 months, *p* = 0.028). The NAT group had a significantly higher R0 rate in those completing surgery (82% vs. 33%, *p* = 0.01).

In 2020, the results of the Dutch PREOPANC-1 phase III RCT were reported, which randomised 246 patients with both resectable and BR-PDAC (based on the Dutch definition) to either upfront surgery or to CRT that consisted of 15 fractions of 2.4 Gy in 3 weeks combined gemcitabine, preceded and followed by a cycle of gemcitabine, with restaging imaging within four weeks before proceeding to surgery within 14–18 weeks from initial randomisation [40]. Overall, 54 (45%) patients in the NAT group and 59 (47%) in the upfront surgery group had BR-PDAC. There was no difference in resection rates between the groups (72/119 (61%) in NAT group and 92/127 (72%) in the upfront surgery group, *p* = 0.58), but the NAT group had a significantly higher R0 rate (71% vs. 40% HR 3.61 (CI 1.87–6.97) *p* < 0.001). There was no difference in median OS between the groups; however, when the subgroup analysis of BR-PDAC alone was performed, it showed significantly improved OS (HR 0.58, CI 0.35–0.95, *p* = 0.029) along with improved DFS for the NAT group. Long-term follow up showed a 5-year OS of 20.5% (CI 14.2–29.8) for the NAT group overall compared to 6.5% (CI 3.1–13.7%) for the upfront surgery group (*p* = 0.025) [41].

The NUPAT-01 trial was a multicentre open-label phase II RCT conducted in Japan, with 51 patients with BR-PDAC (using NCCN criteria) randomised to either neoadjuvant FOLFIRINOX or gemcitabine with nab-paclitaxel (GEM/nab-PTX) and then surgery 2–8 weeks following the last dose of chemotherapy, with a primary endpoint of R0 resection rate [42]. Overall, an R0 resection was achieved in 73% of the FOLFIRINOX group and 56% of the GEM/nab-PTX group (*p* = 0.202), with significantly fewer severe adverse events in the FOLFIRINOX group (30.4% vs. 70%, *p* = 0.01). No difference was seen in 3-year OS between the groups (55.3% and 54.4% respectively, *p* = 0.389).

The multicentre phase II Alliance A021501 RCT in the United States randomised 126 patients with BR-PDAC to either arm 1 (NAT with eight cycles of mFOLFIRINOX) or arm 2 (seven cycles of mFOLFIRINOX and then stereotactic ablative radiotherapy (SABR) (33–40 Gy in five fractions) or hypofractionated image-guided radiotherapy (HIGRT) (25 Gy in five fractions) [43]. Patients without progression then proceeded to resection within 4–8 weeks, which was followed by four cycles of adjuvant FOLFOX6. The primary outcome was 18-month OS, but arm 2 was closed after accruing 56 patients because only 10 of the first 30 patients had successfully undergone R0 resection on interim analysis. The 18-month OS rate was 66.7% (CI 56.1–79.4%) in arm 1 and 47.3% (CI 35.8–62.5%) in arm 2. Ultimately, 38 (58%) patients in arm 1 and 28 patients (51%) in arm 2 underwent surgery, with R0 achieved in 28 (88%) and 14 (74%) patients, respectively. Of note, 82% of patients in the radiation arm received SABR, while 18% received HIGRT, and 16 (50%) of the patients in the SABR group underwent resection, with R0 achieved in 13 of them (81.3%) [44].

The HyperAcute-Pancreas phase III multicentre RCT from the United States randomised patients with BR-PDAC or locally advanced unresectable PDAC (UR-PDAC) based on NCCN criteria to either NAT (FOLFIRINOX or GEM/nab-PTX) plus CRT (50.4 Gy in 28 fractions with either 5-FU or capecitabine) or to NAT including immunotherapy and CRT [45]. In the immunotherapy arm, patients received 3 weekly intradermal injections of 300 million HyperAcute-Pancreas algenpantucel-L (HAPa) immunotherapy cells followed by either FOLFIRINOX or GEM/nab-PTX, during which HAPa was administered every 2 weeks for three more doses, followed by CRT. Overall, 82% of patients had UR-PDAC and 18% BR-PDAC. A total of 24% of patients ultimately underwent resection, with no difference between the arms. OS was 14.9 (12.2–17.8) months in the standard group (*n* = 158) and 14.3 (12.6–16.3) months in the immunotherapy group (*n* = 145) (HR 1.02, CI 0.66–1.58, *p* = 0.98).

The ESPAC 5 study was an open-label phase II RCT conducted between the UK and Germany with 90 patients with BR-PDAC who were randomised to one of four groups: arm 1—immediate surgery, arm 2—neoadjuvant gemcitabine and capecitabine for 2 cycles, arm 3—neoadjuvant FOLFIRINOX for 4 cycles, arm 4—neoadjuvant CRT at a total dose of 50.4 Gy in 28 daily fractions over 5.5 weeks with oral capecitabine [46]. Patients were restaged 4–6 weeks after completing NAT, and those without progression underwent surgery within 2 weeks. A total of 21 (68%) of 31 patients in the immediate surgery group and 30 (55%) of 55 in the pooled NAT groups ultimately underwent resection (*p* = 0.33), while 3 (14%) of 21 patients in the immediate surgery group had an R0 resection, as did 7 (23%) of 30 patients in the combined NAT groups (*p* = 0.49), with the highest proportion in the CRT group (3 of 8 (37%)). No difference was seen in perioperative morbidity between the groups. The 1-year OS was 39% (CI 24–61%) for immediate surgery and 76% (CI 65–89%) for the combined NAT groups (HR 0.29 (CI 0.14–0.60%, *p* = 0.0052)). The 1-year OS was 78% (60–100%) for the neoadjuvant gemcitabine plus capecitabine group, 84% (70–100%) for the neoadjuvant FOLFIRINOX group, and 60% (37–97%) for the neoadjuvant chemoradiotherapy group (*p* = 0.0028).

### Future and Upcoming Trials

The optimum neoadjuvant regime for BR-PDAC has yet to be established, and there are at least 12 ongoing RCTs examining different regimens for both resectable and BR-PDAC, three of which include radiation therapy in at least one treatment arm [47]. This number includes the BPCNCC-1 study, a phase II RCT being conducted in China to compare NAT with GEM/nab-PTX alone to NAT with S1 or GEM/nab-PTX chemotherapy combined with SABR, with the primary outcome being OS [48]. Also included in this number is the Dutch PREOPANC-2 trial, a phase III RCT comparing NAT with eight cycles of FOLFIRINOX to NAT with three cycles gemcitabine, adding hypofractionated radiotherapy (36 Gy in 15 fractions during 3 weeks) to the second cycle and then four cycles of adjuvant gemcitabine post-operatively in patients with both resectable and BR-PDAC [49]. This trial has been completed, and initial reports show equivalent outcomes between the arms in terms of OS, resection rates, and adverse events, but full publication of the results is awaited at time of writing [50]. Similarly, full results are awaited from the completed French PANDAS/PRODIGE 44 RCT, a phase II trial comparing NAT with FOLFIRINOX alone or in combination with CRT for patients with BR-PDAC [51].

## 4. The Role of Radiation Therapy in BR-PDAC

The use of gemcitabine-based CRT has been pursued in BR-PDAC to increase R0 resection and ultimately survival rates, although more aggressive systemic neoadjuvant chemotherapy regimens such as FOLFIRINOX may be adopted without concurrent radiation with a potential survival advantage [52]. A recently published meta-analysis of 31 prospective studies (including 9 RCTs) showed improved R0 resection rates for those patients with BR-PDAC receiving CRT as part of NAT compared to systemic NAT alone (85.2% vs. 62.44%, *p* < 0.0001) [53]. However, concerningly, median OS was shorter in the BR-PDAC NAT-CRT group compared to NAT-alone group (20.84 months (18.14–23.95) vs. 29.38 months (25.01–34.51), *p* = 0.0015). This survival difference does disappear once NAT-CRT is coupled with at least five cycles of induction chemotherapy prior to CRT (23.3 months (14.79–36.73) vs. 21.85 months (13.87–34.42) *p* = 0.8562).

The existing literature is conflicting in terms of the oncological benefit of radiation as part of NAT, not least because of wildly varying NAT regimes and treatment tolerance [54]. The challenge, then, is how to maximise the potential local control of radiation therapy, while minimising interruption to systemic NAT and limiting treatment toxicity [55]. The risk of toxicity delaying or preventing resection must also be considered when incorporating CRT into the neoadjuvant algorithm [56].

## 5. “Conversion Therapy”: NAT for LA-PDAC

Historically, LA-PDAC was considered to be an incurable disease stage and managed with palliative intent, but that paradigm is shifting with the emergence of “conversion strategies” for LA-PDAC, which may allow for surgical resection in a small proportion of patients [57]. Retrospective data from Memorial Sloan Kettering Cancer Center in the United States showed that of 200 patients treated with FOLFIRINOX, 140 went on to radiation therapy and, ultimately, 18% had resection with a median OS of 35.7 months (95% CI 24.3–56.4 months) [58]. Patients who did not undergo resection had a median OS of 20 months (95% CI 16.8–21.9 months). Similar results were seen in a single-centre Spanish study by Roselló et al., which showed a resection rate of 39% and an overall survival of 30.6 months (vs. 13.1 months, *p* < 0.001) for patients with LA-PDAC treated with neoadjuvant FOLFIRINOX and CRT [59]. Of patients with LA-PDAC and a response to FOLFIRINOX or CRT who were listed for resection in Heidelberg, Germany, a resection was achieved in 50.8% of cases, with an R0 rate of 27.3–40.8% and a significantly increased survival compared to those unresectable at surgery [60]. Gemenetzis et al. showed that 20% of patients presenting with LA-PDAC ultimately underwent resection following NAT, with significant survival benefits compared to those not achieving resection (35.3 months (95% CI 24.5–46.0 months) vs. 16.2 months (95% CI 15.2–17.3 months), *p* < 0.001) [61]. Interestingly, they showed the combination of FOLFIRINOX and SABR to be the most likely to be associated with successful resection. A meta-analysis by Brown et al. showed that surgical exploration rates of 41.0% and R0 rates of 16.3% (95% CI 10.7–24.0%) can be seen after NAT for LA-PDAC, with a median survival of 30.0 months (95% CI 20.9–33.0 months) compared to 14.6 months (95% CI 12.8–16.4 months) for those not undergoing resection (*p* < 0.001) [62]. The growing evidence showing a benefit for resection in a minority of patients with LA-PDAC with biologically favourable disease has led to a more aggressive approach for these patients, with studies such as the Dutch PREOPANC-4 trial currently underway evaluating resection rates, perioperative outcomes, and long-term survival for patients undergoing resection following induction chemotherapy [63].

## 6. An Emerging Case for SABR in NAT for BR- and LA-PDAC

SABR allows the safe delivery of high-dose radiation in limited fractions (up to five), allowing ablative dose delivery while reducing the systemic treatment interruption that is seen with traditional CRT [64]. This potentially allows the maximisation of the local control associated with radiotherapy, with more R0 resections and survival advantage, while reducing time off systemic treatment (and time to surgery), reducing the risk of progression [65]. Table 1 summarises key studies published on the use of SABR in the neoadjuvant setting.

The weight of evidence to support the use of SABR in a neoadjuvant setting has come from its role in the setting of unresectable or palliative therapy in PDAC. Herman et al. reported a phase II multicentre study evaluating the feasibility of SABR combined with gemcitabine in LA-PDAC and found minimal evidence of acute or late gastrointestinal toxicity, showing it a feasible treatment strategy [66]. In that study, 8% of patients ultimately underwent R0 resection. In LA-PDAC, a meta-analysis by Tchelebi et al. has shown a survival advantage in patients treated with SABR rather than traditional CRT without surgery, with 2-year OS 26.9% vs. 13.7% (*p* = 0.004), with reduced grade 3/4 acute toxicity (5.6% vs. 37.7% *p* = 0.013) and no difference in late toxicity [67]. In a study examining outcomes for 14,331 patients with unresected PDAC from the National Cancer Data Base in the United States, de Geus et al. showed that treatment with SBRT was associated with a survival advantage when compared to matched patients treated with EBRT (median survival 13.9 vs. 11.6 months, *p* = 0.018) [68]. Similarly, Zhong and colleagues showed SBRT was associated with a significantly improved OS, with an HR of 0.84 (95% CI 0.75–0.93, *p* < 0.001) when compared with matched patients with unresected PDAC treated with EBRT [69].

There is increasing interest in the use of SABR as part of “conversion therapy”—induction chemotherapy, followed by SABR and surgery for LA-PDAC [70]. Quan et al. conducted a phase II trial of 35 patients with BR-PDAC (n = 19) and LA-PDAC (n = 16) treated with gemcitabine/capecitabine induction chemotherapy followed by three-fraction SABR that showed that 34% of the patients ultimately underwent resection, and of those, 92% had an R0 resection [71]. A subsequent phase II trial by Hill and colleagues recruited 44 patients with LA-PDAC and offered induction chemotherapy with mFOLFIRINOX or gemcitabine/nab-paclitaxel prior to five-fraction SABR [72]. A total of 17 (39%) progressed to resection, with an R0 rate of 75%. While no difference was noted in overall survival, successfully undergoing surgery was associated with significantly improved distant-metastases-free survival, HR 0.31 (95% CI 0.12–0.77, *p* = 0.01).

**Table 1 diseases-13-00214-t001:** Details of selected studies reporting outcomes for patients undergoing SABR in neoadjuvant settings.

Study	Year	Design	Patients	Stage	SABR Dose (Gy)	Fractions	Neoadjuvant Chemotherapy	>Gr 3 Toxicity	Number Undergoing Resection (% Total)	R0 Rate (%of Resected)	Major Complications	Adjuvant Therapy (% of Resected Patients)	Median OS (Months)	1-Year OS (%)	Median PFS (Months)	1-Year PFS (%)
**Alliance A021501** [43]	2022	Phase 2 RCT: Arm 1 NAT with mFOLFIRINOX 8#; Arm 2 NAT with mFOLFIRINOX 7# plus SABR or HIGRT	Arm 1: 65; Arm 2: 55	BR-PDAC	33–40	5	mFOLFIRINOX 7#	Arm 1: 37 (57%); Arm 2: 35 (64%)	Arm 1: 30(46%); Arm 2: 18 (33%)	Arm 1: 28 (93%); Arm 2: 14 (78%)	NA	4# FOLFOX6 Arm 1: 22 (69%); Arm 2: 13 (69%)	Arm 1: 29.8 (21.1–36.6); Arm 2: 17.1 (12.8–24.4)	18-month OS— Arm 1: 66.7 (56.1–79.4); Arm 2: 47.3 (35.8–62.5)	Event-free Survival—Arm 1: 15.0 (11.2–21.9); Arm 2: 10.2 (6.7–17.3)	NA
**Herman et al.** [66]	2014	Phase 2 Multicentre Trial	49	LA-PDAC	33	5	Gemcitabine induction	7 (14%)	4 (8%)	4 (100%)	NA	NA	13.9 (10.2–16.7)	59%	7.8 (5.8–10.2)	32%
**Quan et al.** [71]	2018	Phase 2 Single-Centre Trial	35	BR-PDAC (54%) LA-PDAC (46%)	36	3	Gemcitabine/Capecitabine 4#	0	12 (34%)	11 (92%)	4(33%)	Multiple regimens	18.9 (12.8–30.9)	NA	22.3 (12.4–28.1)	NA
**Hill et al.** [72]	2022	Phase 2 Multicentre Trial	48	LA-PDAC (44) Locally recurrent (4)	25–33	5	mFOLFIRINOX 31 (65%), GnP 15 (31%), other 8 (16%)	13 (27%)	16 (33%)	12 (75%)	NA	Not proscribed	14.6 (11.6–23)	58%	6.4 (5.0–12.7)	
**Zakem et al.** [73]	2021	Single-Centre Retrospective Cohort	103	BR-PDAC 85(83%) LA-PDAC 18(17%)	30–33	5	FOLFIRINOX 69 (67%), GnP 28 (27%), other 6 (6%)	0	73 (71%)	71 (97%)	6 (8%)	Gemcitabine based (53%); FOLFIRINOX (14%)	24	NA	NA	NA
**Palm et al.** [74]	2023	Single-Centre Retrospective Cohort	303	BR-PDAC	40 (30–55)	5 (105)	GTX (48%) FOLFIRINOX (29%) GnP (17%)	10 (3%)	169 (56%)	155 (92%)	NA	Not proscribed	25 (22–28)	NA	NA	NA
**Blair et al.** [75]	2018	Single-Centre Retrospective Cohort of Resected Patients	168 (61 (36%) NAT with SABR; 107 (64%) NAT with CRT)	BR-PDAC 84 (50%) LA-PDAC 84 (50%)	33	5	FOLFIRINOX 70 (42%) Other multiagent 69 (41%)	NA	168 (100%)	135 (80.4%)	44 (26%)	NA	NA	NA	NA	NA
**Song et al.** [76]	2023	Single-Centre Retrospective Cohort of Resected Patients	62	Resectable 18 (29%); BR-PDAC 30 (48.4%); LA-PDAC 14 (22.6%)	52.6 (30.8–60.8)	5	FOLFIRINOX 57 (92%) GnP 2 (3%) None 3 (5%)	NA	62 (100%)	52 (85%)	11 (18%)	Gemcitabine based (42%); FOLFIRINOX (34%); Other (16%); None (8%)	NA	95.50%	NA	93.1% (LRFS)

# = Number of chemotherapy cycles, NA = not applicatble.

Some single-institution reports show R0 rates of >90% in those undergoing resection following NAT, including SABR for BR-PDAC [73,77]. Despite these promising findings, the early closure of the SABR arm of the Alliance A021501 study due to low the R0 rate on interim analysis remains a potential concern, even if the SABR group ultimately had a 50% resection rate and 81.3% R0 rate as described above [43,44]. Interestingly, Palm et al. reported a single-centre study that found that a positive margin, defined in their study as a margin ≥ 0 mm (“no tumour on ink”), had no impact on long-term oncological outcomes for patients treated with neoadjuvant chemotherapy and SABR, potentially suggesting the impact of an R1 margin following ablative radiotherapy may be less important than in other treatment regimens [74]. Further attention will be required going forward to address this question. It is also true, however, that emerging surgical techniques such as total mesopancreatic excision (TMpE), which includes complete dissection of the mesopancreatic lamina along the major retroperitoneal vessels, may allow improvements in the R0 rate [78]. A meta-analysis by da Silva et al. showed that TMpE significantly improved the R0 rate compared to standard pancreaticoduodenectomy (RR 1.24 (1.11–1.38) *p* < 0.05) without increasing perioperative morbidity and led to a reduction in both the local and overall recurrence rates [79].

## 7. The Impact of EBRT on Perioperative Outcomes Following Pancreaticoduodenectomy

Pancreaticoduodenectomy is associated with significant risks of perioperative morbidity, the rates of which have been estimated from 30 to 60% [80]. Historically, pancreaticoduodenectomy was associated with a perioperative mortality risk of 25–30%, but the risk is generally reported as <5% in contemporary studies [81]. Cools et al. utilised data from the American College of Surgeons-National Surgical Quality Improvement Project (ACS-NSQIP) to evaluate the impact of NAT on perioperative outcomes for 3748 patients undergoing pancreaticoduodenectomy, showing no difference in overall complications (55.8% vs. 55.1%, *p* = 0.685) or major complications (16.7% vs. 18.3%, *p* = 0.287) when compared with patients having upfront surgery [82]. Similarly, Aploks et al. evaluated perioperative outcomes for patients undergoing pancreaticoduodenectomy following CRT compared to upfront surgery and showed that following CRT, operative time increased (OR 1.114 (1.11–1.16, *p* < 0.001)) and there was an increased need for perioperative transfusion (OR 1.58 (1.37–1.82, *p* < 0.001)), potentially suggesting more difficult resection [56]. This corresponds to the experience of surgeons, who are more likely to classify a pancreaticoduodenectomy as difficult following NAT [83]. Interestingly, NAT with CRT has been shown to change the composition of the pancreas, leading to a stiffer gland [84]. One advantage of this is that it reduces the risk of overall post-operative pancreatic fistula (POPF) (OR 0.59 (0.47–0.74, *p* < 0.001) and the risk of clinically relevant POPF (OR 0.45 (0.37–0.53, *p* < 0.001), without a difference in the overall risk of perioperative morbidity [85].

## 8. Pancreaticoduodenectomy Following SABR

Limited published data exist on the perioperative outcomes for patients undergoing pancreatic resection post-SABR, but a single-centre series by Blair et al. from the United States found no increased risk of perioperative morbidity when compared with patients who underwent neoadjuvant CRT, with major complications occurring in 23% and 28% of patients, respectively (*p* = 0.471) [75]. A single-centre series including 61 patients undergoing resection following NAT including SABR from Florida, USA, by Mellon et al., found that 39% of patients experienced post-operative complications, including 10 patients (16%) who had a POPF or bile leak [86]. Similarly, a single-centre series including 62 patients from Seoul, Republic of Korea, which included patients undergoing both pancreaticoduodenectomy and distal pancreatectomy, found that 29% of patients experienced complications, including 11 (18%) who experienced major morbidity [76].

### 8.1. SABR and Patient-Reported Outcomes

As described earlier, SABR has emerged into the setting of NAT having first been utilized for local tumour control in a palliative setting. Koong et al. showed that SABR could be highly effective in achieving symptom control for those patients with metastatic PDAC and significant abdominal pain [87]. The SABR-5 trial was a phase II study examining the role of SABR in the treatment of oligometastases and incorporated a quality of life (QoL) assessment that found that while patients had transient worsening of QoL during SABR, the majority achieved long-term stable QoL when compared to pre-treatment [88]. The Dutch PANCOSAR RCT will evaluate this further, comparing OS and QoL outcomes for patients with localized PDAC who are unfit for surgery or systemic therapy who are randomized to either palliative SABR or best supportive care [89].

Interestingly, the Alliance A021501 study collected patient-reported outcomes (PROs) for adverse events and found a significant QoL decrease for patients undergoing NAT [90]. Patients in arm 2 (chemotherapy plus SABR/HIGRT) reported significantly worse appetite than those in arm 1 (chemotherapy alone), but no other significant differences were noted. In their single-arm phase II trial, Herman et al. noted a stable QoL reported by patients undergoing SABR for LA-PDAC, with a significant improvement in pancreatic pain [66]. Similarly, SABR appears to have been well tolerated by patients in the phase II trial conducted by Quan et al., with no change in patient-reported QoL and no recorded Grade 3 toxicity for patients with BR- and LA-PDAC [71].

### 8.2. Limitations in the Evidence for SABR

At present, the evidence to support the use of SABR in the setting of NAT for PDAC is limited. The most significant study on it to date, the multicentre phase II Alliance A021501 RCT, closed arm 2 at interim analysis due to failing to meet a pre-defined minimum of 11 of the first 30 patients undergoing R0 resection [43]. As a result, the failure to meet full accrual hampers the generalizability of the results, as does the confounding of two distinct radiotherapy approaches in arm 2. Equally, however, much of the supporting data for SABR is inferred from registry data or from its role in LA-PDAC, with single-centre retrospective evidence informing practice in the NAT setting [70]. The majority of prospective evidence to date comes from the United States, but the anticipated BPCNCC-1 study from China will help to address this imbalance.

## 9. Conclusions

The evidence for NAT in the setting of BR-PDAC is compelling, but the optimum regime is yet to be established and the role for radiation therapy is still under debate. With a large number of RCTs currently underway, or awaiting full publication, no doubt the paradigm will evolve going forward. The potential advantages of SABR are in limiting toxicity and interruptions to systemic treatment while giving local tumour control; however, data to support its use in the setting of BR-PDAC are conflicting. Despite this, it has recently come into use in the setting of BR-PDAC in our institution. Clearly, tumour biology and careful patient selection are critical to optimizing the benefit of SABR. Long-term oncological outcomes will need to continue to be evaluated as more data become published; indeed, the argument for NAT with chemotherapy alone may ultimately be the most compelling. In the case of LA-PDAC, the rationale for the utilisation of SABR as part of a neoadjuvant conversion strategy is stronger, with significant survival benefit to the minority of patients who successfully undergo resection.

## Data Availability

No additional data created.

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
