# Peer review of "Neoadjuvant Stereotactic Ablative Radiotherapy in Pancreatic Ductal Adenocarcinoma: A Review of Perioperative and Long-Term Outcomes"

_diseases, 2025, doi:10.3390/diseases13070214_

Round 1
Reviewer 1 Report
Comments and Suggestions for Authors
This review disscusses the role of neo-adjuvant radiochemotherapy, particularly SABR, in the management of borderline and locally advanced pancreatic carcinoma against upfront surgery and palliative treatment, repectively. It is informative. Addition of a table summarizing relevant hitherto data may be useful.
Author Response
Comment: Addition of a table summarizing relevant hitherto data may be useful
Response:
Thank you for your helpful suggestion. We have included a summary table as a supplemental file.
Reviewer 2 Report
Comments and Suggestions for Authors
The article is well organized and writed, reporting many correct data and good references. The value and efficacy of neodaiuvant therapy in reducing the R1 resection in borderline resectable PADC, has been higligheted. What should be stressed are the real incidence of complications both after neoadjuvant chemotherapy that reduces the number of patients who then go to surgery, and the complications resulting from neoadiuvant radiotherapy, even short term. The evidence of these data would make it more justifiable to indication to neoadjuvant radiotherapy, not only in locally advanced cases, as reported in the article, but also in borderline resectable tumors. Last topic to discuss, during the Pancreatoduodenectomy, is the role of Total Mesopancreas Excision to obtain the complete clear of posterior margin of pancreas frequent site of R1. Without the application of this technical procedure the advantage of neoadjuvant chemo-RT are to be considered doubtful
Author Response
Comment 1. What should be stressed are the real incidence of complications both after neoadjuvant chemotherapy that reduces the number of patients who then go to surgery, and the complications resulting from neoadiuvant radiotherapy, even short term.
Response 1. Thank you for highlighting this point, and we certainly agree treatment toxicity must be factored into the equation. We have expanded the manuscript to include data from Labori et al., 2024 and Aploks et al., 2023 to emphasise the point for chemotherapy and radiotherapy respectively.
Comment 2. Last topic to discuss, during the Pancreatoduodenectomy, is the role of Total Mesopancreas Excision to obtain the complete clear of posterior margin of pancreas frequent site of R1
Response 2. This is an interesting point, and certainly merits inclusion. We have incorporated data from Safi et al, 2021, and a metanalysis by da Silva et al., 2024 to expand on this point.
Reviewer 3 Report
Comments and Suggestions for Authors
Thank you for the opportunity to review this important manuscript. Here are my comments and suggestions.
Remove the complete segment of ''Epidemiology of PDAC'' except the last paragraph. Not important for the discussion about resectability.
Line 102: define ''tumor biology''. Also, is it essential for the definition of borderline or advanced local disease? Is tumor biology included in the guidelines or recommendations, including NCCN?
Line 129: remove free space after ''CRT''
Line 131: replace ''neoadjuvant therapy'' with ''NAT''. Check in the remaining manuscript.
Line 140: remove free space before the closing bracket.
In the part ''The Evidence for Neoadjuvant Therapy in BR-PDAC'' the period between neoadjuvant therapy and surgery should be mentioned.
Line 223: Replace ''neoadjuvant chemoradiation regimens '' with ''NAT''. It shortens the text.
Line 227: Replace ''metanalysis'' with ''meta-analysis''.
Line 252: Replace ''at time of surgery'' with ''at surgery''.
Line 257: Replace ''metanalysis'' with ''meta-analysis''.
Line 270: Do authors mean ''with more R0 resections''.
Line 279: Replace ''metanalysis'' with ''meta-analysis''.
Line 282: remove free space after the closing bracket.
The main point here is the results of the study mentioned in the text ''In LA-PDAC, a metanalysis by Tchelebi et al., has shown a sur-279 vival advantage in patients treated with SABR rather than traditional CRT without sur-280 gery, with 2-year OS 26.9% v 13.7% (p=0.004), with reduced grade 3/4 acute toxicity (5.6% 281 v 37.7% p=0.013) and no difference in late toxicity''. Is there a need for surgery after successful SBRT in LA-PDAC? The survival is comparable.
Lines 282-287 have a different font.
Author Response
Remove the complete segment of ''Epidemiology of PDAC'' except the last paragraph. Not important for the discussion about resectability.
We respectfully disagree with the reviewer on this point, it is our intention to try and build the argument for neoadjuvant SABR by placing it within the context of PDAC as a whole – first by introducing it, then by discussing the classification of resectability, then by discussing the evidence for NAT, and finally introducing the concept of SABR as a logical step from what has gone before.
Line 102: define ''tumor biology''. Also, is it essential for the definition of borderline or advanced local disease? Is tumor biology included in the guidelines or recommendations, including NCCN?
We have amended that statement to read “worse tumour behaviour” as that may be a more clear statement of what we intend. We have amended the following sentence to clarify that this is not considered in the NCCN guidelines, but it is included in the IAP consensus statement which we go on to discuss in the following sentences.
Line 129: remove free space after ''CRT''
Done.
Line 131: replace ''neoadjuvant therapy'' with ''NAT''. Check in the remaining manuscript.
Thank you for highlighting – four replacements made in the manuscript.
Line 140: remove free space before the closing bracket.
Done thank you.
In the part ''The Evidence for Neoadjuvant Therapy in BR-PDAC'' the period between neoadjuvant therapy and surgery should be mentioned.
We have updated the discussion of each trial to include the interval to surgery.
Line 223: Replace ''neoadjuvant chemoradiation regimens '' with ''NAT''. It shortens the text.
Done.
Line 227: Replace ''metanalysis'' with ''meta-analysis''.
Done, and all other references to metanalysis similarly changed.
Line 252: Replace ''at time of surgery'' with ''at surgery''.
Done
Line 257: Replace ''metanalysis'' with ''meta-analysis''.
Done.
Line 270: Do authors mean ''with more R0 resections''.
We do, thank you for pointing this out. We have changed the manuscript.
Line 279: Replace ''metanalysis'' with ''meta-analysis''.
Done.
Line 282: remove free space after the closing bracket.
Done.
The main point here is the results of the study mentioned in the text ''In LA-PDAC, a metanalysis by Tchelebi et al., has shown a sur-279 vival advantage in patients treated with SABR rather than traditional CRT without sur-280 gery, with 2-year OS 26.9% v 13.7% (p=0.004), with reduced grade 3/4 acute toxicity (5.6% 281 v 37.7% p=0.013) and no difference in late toxicity''. Is there a need for surgery after successful SBRT in LA-PDAC? The survival is comparable.
That’s an interesting point, but we would argue that this merely shows the advantage of SABR over CRT. In the subsequent paragraph, we highlight data from a phase II study by Hill et al. that shows those patients with LA-PDAC who ultimately undergo resection after SABR have improved progression free survival compared to the SABR arm alone.
Lines 282-287 have a different font.
Thank you for pointing that out, it has been corrected